# Selenium Combined with Methyl Jasmonate to Control Tomato Gray Mold by Optimizing Microbial Community Structure in Plants

**DOI:** 10.3390/jof8070731

**Published:** 2022-07-14

**Authors:** Changyin Li, Chengxiao Hu, Jiatao Xie, Guangyu Shi, Xu Wang, Xiang Yuan, Keyi Li, Siqi Chen, Xiaohu Zhao, Guocheng Fan

**Affiliations:** 1State Key Laboratory of Agricultural Microbiology, College of Resources and Environment, Huazhong Agricultural University, Wuhan 430070, China; llcy@webmail.hzau.edu.cn (C.L.); hucx@mail.hzau.edu.cn (C.H.); jiataoxie@mail.hzau.edu.cn (J.X.); yxhzau@163.com (X.Y.); shuangsiwang2@163.com (K.L.); csq18120210718@163.com (S.C.); 2Fujian Key Laboratory for Monitoring and Integrated Management of Crop Pests, Fuzhou 350013, China; 3College of Environment Science and Engineering, Suzhou University of Science and Technology, Suzhou 215009, China; shigy@usts.edu.cn; 4Institute of Quality Standard and Monitoring Technology for Agro-Products of Guangdong Academy of Agricultural Sciences, Guangzhou 510640, China; wangxuguangzhou@126.com

**Keywords:** tomato, gray mold, incidence, Selenium, Methyl jasmonate, microbes

## Abstract

Tomato cultivation is seriously affected by infection from *Botrytis cinerea*. The safe and effective control of tomato gray mold remains elusive. Plant-related microbial communities regulate not only plant metabolism but also plant immune systems. In this study, we observed that Selenium application in soil combined with foliar spraying of methyl jasmonate could reduce *Botrytis cinerea* infection in tomato fruits and leaves and improve tomato fruit quality. The infection rate of leaves decreased from 42.19% to 25.00%, and the vitamin C content increased by 22.14%. The bacterial community structure of the tomato was studied by using amplicon sequencing technology. The leaf bacterial alpha diversity of tomatoes treated with Se plus methyl jasmonate was significantly higher than that of the control. Then we isolated five strains antagonistic to *Botrytis cinerea* in vitro from tomato leaves in the treatment of Se plus methyl jasmonate. The antagonistic strains were identified as *Bacillus subtilis* and *Bacillus velezensis*. Spraying mixed antagonistic strain suspension significantly inhibited the diameter of *Botrytis cinerea* with an inhibition rate of 40.99%. This study revealed the key role of plant-beneficial bacteria recruited by Se combined with methyl jasmonate in improving tomato plant disease resistance. These findings may benefit our understanding of the new regulation of microorganisms on *Botrytis cinerea*.

## 1. Introduction

The tomato (*Lycopersicon esculentum*) is considered to be one of the most important vegetables in the world. Because the tomato fruit is succulent and has a thin peal, it is easily infected by pathogens, thus causing disease. *Botrytis cinerea* is a pathogen that causes gray mold; it can resist an adverse environment by producing spores and sclerotia, thus infecting the host and causing disease, even in the postharvest storage period [1]. This pathogen causes the tomato to become moldy and lose its commercial value during transportation and storage [2]. The global economic loss caused by gray mold is as high as $10 billion a year. At present, the prevention and control of gray mold depends mainly on chemical agents [3]. However, the overuse of fungicides endangers the safety of agricultural products and leads to the enhancement of the drug resistance of pathogenic fungus and to environmental pollution and ecological imbalance, thus causing the product to fail to meet the requirements of sustainable development of modern agriculture [4]. Therefore, it is of great importance to find and develop ecological control technologies for gray mold.

Plants recruit abundant and specific microorganisms that play important roles in plant life [5]. Beneficial microorganisms in plants can promote plant growth [6], absorb nutrients [7], tolerate adversity [8], and resist pathogens [9]. Generally, healthy plants exhibit high diversity of microbial communities and good connectivity of microbial networks [10]. Plant-related microbial communities are formed by the interaction of the host [11], microorganisms, environmental factors [12], and agronomic measures. More and more studies have found that plants can resist the invasion of pathogens by recruiting beneficial microorganisms and regulating microbial community structure [13]. For example, a study by Liu et al. [14] has shown that after wheat was infected by *Fusarium pseudograminearum*, *Stenotrophomonas rhizophila* was enriched in the rhizosphere and roots, and this microorganism can protect the wheat by promoting plant defense ability. In modern agricultural practice, manipulating plant microorganisms to resist pathogens is considered to be a promising method [15].

Selenium (Se) is an essential trace element for the human body and a beneficial element for plants. Studies showed that suitable concentration of Se can improve the quality and yield of crops [16,17] and enhance crop antioxidant capacity and photosynthesis [18,19]. In addition, Se has a good effect on alleviating plant stresses such as heavy metal stress [20], salt injury [21], freezing injury [22], high temperature [23], drought [24], and disease [25]. It has been reported that Se significantly reduced the incidence of apple rot [26], tomato Fusarium wilt [27], and oilseed rape Sclerotinia [25] during plant growth. Our previous studies have shown that Se can optimize the rhizosphere soil microbial community and enhance the resistance of oilseed rape to *Sclerotinia sclerotiorum* [28]. Even though Se was shown to have a positive effect on plant health, Se is arguably a micro-element for biology with a very narrow window between deficiency and toxicity when compared with other trace elements. Therefore, it was necessary to control the application concentration of Se to make sure it was safe for plants [20,25].

Methyl jasmonate (MeJA) is a growth regulator widely found in plants. Because of its nontoxicity and its high level of safety to the environment, it has often been used in recent years to replace chemical fungicides in fruit and vegetable disease prevention. Findings showed that appropriate concentration of MeJA could improve the disease resistance of tomatoes [29], bayberries [30], and mandarins [31] and thus reduce the fruit rot caused by pathogens.

Based on the biological functions of Se and MeJA, as well as their application to the prevention and control of crop diseases, we hypothesized that the combination of Se and MeJA might have a synergistic effect in preventing gray mold disease. However, there was no research on the combined application of Se and MeJA for controlling crop diseases. The intent of this study was to use a combination of experimental approaches to identify the effects of combined application of Se and MeJA on the control of tomato gray mold and contributory microbiological mechanisms. First we studied the combined effects of Se and MeJA on the control of the disease, and then we analyzed the effect on tomato fruit quality. Next, on the basis of the results obtained, we analyzed the microbiome profiles of tomato fruits and leaves under the treatment of Se combined with MeJA. Moreover, several strains that showed significant antagonistic effects against *Botrytis cinerea* under in vitro culture conditions were isolated and obtained. Finally, the obtained antagonistic strains were inoculated into live tomato plants to study their prevention effects against gray mold disease. The study will help to clarify the prevention and control effect of Se combined with MeJA on tomato gray mold and reveal its prevention and control mechanism from the point of view of microorganisms.

## 2. Materials and Methods

### 2.1. Experimental Design

We chose for this study the tomato (*Solanum*) variety *Micro Tom*, obtained from Baker Creek. Pot experiments were conducted at Huazhong Agricultural University, Wuhan, China. We transplanted one tomato seedling per pot. The basic physical and chemical properties of the soil in the pots were as follows: organic matter 13.72 g/kg, pH 7.12, available hydrolysis N 52.33 mg/kg, available P16.57 mg/kg, available K 193.48 mg/kg, and total Se 0.092 mg/kg.

Three experiments (*experiments I to III*) were set up: 

***Experiment I*** started on 1 September 2020. Se was added to the soil in the form of Na_2_SeO_3_. After mixed well, 3 kg of soil was added into each pot. MeJA was sprayed on the surface of the tomatoes at 12, 13, and 14 weeks after planting. Each pot of tomato was sprayed with a total 15 mL. The design of the experiment was as follows: (1) Control (CK); (2) 0.3 mg/kg Se (Se0.3); (3) 2.0 mg/kg Se (Se2.0); (4) 0.2 mmol/L MeJA (MeJA); (5) 0.2 mmol/L MeJA + 0.3 mg/kg Se (MSe0.3); (6) 0.2 mmol/L MeJA + 2.0 mg/kg Se (MSe2.0). Each treatment comprised four replicates. Here the dose of Se was selected based on our previous studies and on the soil Se concentration grade [20,25,28].

***Experiment II*** started on 1 April 2021. In order to further verify the inhibitory effect of Se and MeJA on *Botrytis cinerea* of tomato, 2.0 mg/kg of Se was mixed into the soil and 0.2 mmol/L MeJA was sprayed after 7 weeks of tomato growth. *Botrytis cinerea* was inoculated at the end of the eighth week. Each treatment comprised eight replicates.

***Experiment III*** started on 1 September 2021. In order to verify the control effect of the antagonistic bacteria on *Botrytis cinerea* of tomato, bacterial suspension—of which the selected strains were isolated from the tomato leaves in MSe2.0 treatment of ***Experiment I***—was sprayed on the leaves for 7 weeks starting from the time the tomato was planted. The experiment included four treatments: (1) Control (CK); (2) the plants were sprayed with 10 mL *Bacillus velezensis* suspensions (OD_600_ = 1) (BV); (3) the plants were sprayed with 10 mL *Bacillus subtilis* suspensions (OD_600_ = 1) (BS); (4) the plants were sprayed with 5 mL *Bacillus velezensis* suspensions and 5 mL *Bacillus subtilis* suspensions (OD_600_ = 1) (BVS). Two days after spraying the beneficial bacteria, the *Botrytis cinerea* was inoculated. Each treatment comprised eight replicates.

### 2.2. Determination of Disease Incidence

#### 2.2.1. Determination of the Disease Incidence of Tomato Fruits

According to the method described by Wang et al. [32], the fruits were wounded using a sterile needle from each fruit, then inoculated with 5 μL spore suspension (1 × 10^5^ spores/mL) of *Botrytis cinerea* at each wound. After drying, the tomatoes were placed in trays and stored at (23 ± 1) °C. After 72 h, the grey mold severity was evaluated using a scale of 0 to 4 (0: no spots; 1: 0–3 mm; 2: 3–6 mm; 3: 6–9 mm; 4: >9 mm) based on the diameter of the lesions on the tomato surface (Figure 1E). All the treatments were repeated—eight replicates, with 20 tomato fruits per replication.

#### 2.2.2. Determination of the Disease Incidence of Tomato Leaves

The disease incidence of tomato leaves was determined according to two methods. (1) ***Hyphae infection***: the tomato leaves were inoculated *Botrytis cinerea* mycelia plugs by using a sterile toothpick. Mycelia plugs were placed between the main leaf vein and the leaf edge and gently pressed to make it fit with the leaf. All the leaves were same-size- wounded with a sterile needle tip before inoculation. After 48 h, the lesion sizes were measured on each leaf for every treatment. Each treatment contained four replicates, and 4 leaves were selected for each repetition [25]. (2) ***Spore infection***: using spore suspension spray inoculation method, 10 mL spore suspension (1 × 10^5^ spores mL^−1^) of *Botrytis cinerea* was evenly sprayed on the tomato seedlings under different treatment. Each seedling scratched 16 leaves with a sterilized blade. The incidence of *Botrytis cinerea* on the leaves was investigated 7 days after inoculation with pathogenic fungi.

### 2.3. Determination of the Contents of Se, Soluble Protein, and Vitamin C

The Se content in tomato leaves and fruits was determined using hydride atomic fluorescence spectrometry (GB 5009.93-2010). We weighed a 0.3 g sample in the digestion flask, then added 10 mL HNO_3_-HClO_4_ (9:1) of mixed acid and heated it on a hot plate. After it heated to 2 mL, we cooled it down and added 5.0 mL 6 M hydrochloric acid. Finally, the solution became clear and colorless. We transferred the digestion solution to a volumetric flask, added concentrated hydrochloric acid, and measured the Se content with an atomic fluorescence spectrometer. The soluble protein and vitamin C were determined according to the *Plant Physiological and Biochemical Test Technology and Principle (Version 2)*. The soluble protein was determined by the Coomassie Brilliant Blue method. The vitamin C was determined by the colorimetric method using 2,6 dichlorophenol.

### 2.4. Isolation of Bacteria from Tomato Leaves

The bacteria were isolated by the plate dilution method according to Fan et al. [33] with minor modifications. Ground and homogenized tomato leaves with MSe2.0 treatment (of ***Experiment I***) were diluted to 10^−1^–10^−5^ concentration gradient, 100 μL was coated on Beef Extract Peptone Agar Medium plate, incubated at 30 °C for 24 h, and the colonies with different morphology and color were purified then stored at 4 °C.

### 2.5. Determination of the Antagonistic Activities of Bacteria against Botrytis Cinerea In Vitro

The antagonistic strains were screened by the plate confrontation method, as described by Yin et al. [34]. The *Botrytis cinerea* and 15 bacterial strains isolated from the plate were inoculated on the PDA plate to carry out the confrontation test, and the bacteria with the bacteriostatic zone were screened out.

### 2.6. DNA Extraction and Sequencing

#### 2.6.1. Identification of Antagonistic Strains

A Mabio Bacterial DNA Kit (DNB361-03B) was used to extract the genomic DNA of the target strain. PCR amplification was performed using 16S universal primers 27F (AGAGTTTGATCCTGGCTCAG) and 1492R (TACGGYTACCTTGTTAYGACTT). The amplified products were sequenced by an ABI3730 sequencing platform. The sequencing results were Blast-compared with the NT Library of NCBI to obtain the species annotation classification results.

#### 2.6.2. 16S rRNA Amplicon Sequencing

The genomic DNA from the tomato leaves and fruits was extracted and purified by using a MOBIO PowerSoil DNA Isolation Kit (MOBIOLaboratories, Carlsbad, CA, USA) following the manufacturer’s recommendations [8]. The universal primers 515F and 806R were used for the amplification of the bacterial 16S rRNA genes in the leaf samples. PCR products were examined using agarose gel electrophoresis. Using E. Z. n. a^®^ GelExtraction Kit Gel Recovery Kit, we recovered PCR mixed products and TE buffer eluted and recovered target DNA fragments. Finally, PCR products were sequenced on the Illumina Hiseq platform. The high-quality sequences were analyzed using the standard operating procedure in QIIME.

### 2.7. Statistical Analysis

All statistical analyses were performed in the SPSS software (v22.0) or R (v3.6.2) environment. One-way analysis of variance (ANOVA) was carried out to analyze the results, which included the visual score of *Botrytis cinerea,* Se, vitamin C, and the soluble protein content in the tomatoes. The independent T-test was used to analyze the incidence of tomato leaves between the two groups. The “vegan” package in R based on the Bray–Curtis dissimilarity was used to analyze the PCoA. The linear discriminant analysis (LDA) effect size (LEfSe) was performed to analyze the relative abundance of bacterial taxa between treatments [35].

## 3. Results

### 3.1. Evaluation of Se andMeJA for the Prevention and Inhibition of Botrytis cinerea in Tomato Fruits

As indicated in Figure 1A, three days after inoculation with *Botrytis cinerea* spores, the lesion size of the tomato fruits treated with Se0.3, Se2.0, MSe0.3, and MSe2.0 was significantly smaller than that of the control group. In order to determine the exact inhibitory effect of Se and MeJA on *Botrytis cinerea* of tomatoes, grey mold severity was evaluated using a scale of 0 to 4 (0: no spots; 1: 0–3 mm; 2: 3–6 mm; 3: 6–9 mm; 4: >9 mm) based on the diameter of the lesions on the tomato surface (Figure 1E). The average score of the CK group was 32, while the average score of the MSe2.0 treatment group was 15.75 (each repeat picking 20 fruits) (Figure 1D). When the tomato leaves were inoculated against *Botrytis cinerea* by using a sterile toothpick with an agar plug, the average diameter of the lesions was 1.28 cm and the inhibition rate was 7.24% (Figure 1C). When the tomato seedlings were inoculated with a spore suspension spray, the infection rate of the leaves decreased dramatically, from 42.19% to 25.00% (Figure 1B). The combined effect of Se and MeJA on the control of postharvest gray mold of the tomato was better than Se or MeJA applied alone. Overall, the combined treatment of Se and MeJA significantly inhibited the infection and expansion of *Botrytis cinerea*.

### 3.2. Effects of Se and MeJA on Se Content, Vitamin C Content, and Soluble Protein Content in Tomatoes

With the increase in Se application in soil, the absorption of Se in the tomato fruit and leaf increased. The Se content of the tomato leaves was 151.30 μg/kg DW in the control group, and the Se2.0 treatment was 5.24 times higher than in the control group (Figure 2A). The Se content of the tomato fruits was 27.48 μg/kg DW in the control group and 329.93 μg/kg DW in the Se2.0 treatment group (Figure 2B).

The vitamin C and soluble protein content were identified to assess the quality of the tomato fruits. The vitamin C and soluble protein content of the tomato fruits increased after applying Se in the soil and/or foliar spraying MeJA. MSe2.0 treatment increased the vitamin C content of the tomato fruits by 22.14% when compared with the control (Figure 2C). When 2 mg/kg Na_2_SeO_3_ was applied in the soil combined with MeJA of 0.2 mmol/L (MSe2.0), the content of the soluble protein was the highest (6.84 mg/g) among the treatments, namely 43.37% higher than that of the control group (Figure 2D).

### 3.3. Effect of Se and MeJA on the Bacterial Community Composition and the Diversity-Inhabiting Fruits and Leaf of the Tomato

A total of 84,467 to 92,385 sequences were obtained from the 16S (Bacteria) regions of the leaf and fruit of the tomato. After cleaning the data, 85,725 to 91,558 16S reads were retained. In order to visualize the differences in the bacterial community compositions on a class and order level between the control group and the MSe2.0 treatment, the relative abundance of bacteria in the top 15 most abundant groups is shown in Figure 3. Among the bacterial class in the tomato leaf, *Gammaproteobacteria* dominated heavily with 70%, followed by *Bacteroidia* (11%), *Alphaproteobacteria* (10%), and *Actinobacteria* (3%). *Bacilli* was only detected in tomato leaves in the MSe2.0 treatment (Figure 3A,B).

By comparing the alpha and beta diversity of microbiota in the tomato leaf and fruit, the leaf bacterial Chao-1 index estimates revealed the tomato leaves of MSe2.0 treatment (L-MSe2.0) to harbor more richness microbiota than the control group (Figure 3C). The leaf bacterial Shannon diversity index in the MSe2.0 treatment was significantly higher than in the control group (Figure 3D). The PCoA of the Bray–Curtis distance showed that the distinct structure of the bacterial phyllosphere microbiota in the MSe2.0 treatment formed separated clusters in the first two coordinate axes (Figure 3E). All of the profiles indicated that the tomato phyllosphere microbiota changed with the Se and MeJA treatment. 

### 3.4. Effect of Se and MeJA on Peculiar Clades among the Bacterial Communities of Tomato Leaf

The linear discriminant analysis effect size (LEfSe) detected 26 bacterial species in the leaf (LDA scores > 4), which discriminated the bacterial communities between treatment of CK and MSe2.0 (Figure 4A). *Rhodanobacter* was the most influential bacterium in the leaves receiving MSe2.0 treatment. Compared with the control, tomato leaves treated with Se and MeJA have more opportunities to recruit beneficial bacteria. A cladogram (Figure 4B) indicated the taxonomic component of bacterial microbial communities; it can better visualize the change in bacterial community in phyllosphere from class to genus under MSe2.0 treatment. *Bacilli* (from class to genus), *Chitinophagales* (from class to genus), and *Bateproteobacteriales* (from class to genus) were significantly enriched in MSe2.0 treatment.

### 3.5. Isolation of Phyllosphere Bacteria and Their Antagonism with Botrytis cinerea

The tomato leaf samples with the lowest incidence rate of MSe2.0 treatment were collected, and the microbes were separated and purified. A total of 15 bacteria were isolated according to the appearance and color characteristics of the colony (Figure 5A). These bacteria were then tested in confront culture assays for their antagonism against *Botrytis cinerea*. Five of the 15 bacterial species exhibited antagonistic activities to *Botrytis cinerea* and formed inhibition zones (Figure 5B). The spliced sequences were compared with the NT library of NCBI by blast to obtain the annotation classification results; these were *Bacillus velezensis* and *Bacillus subtilis* (Figure 5C). They were used for the subsequent study because they exhibited antagonistic activities to the pathogen.

### 3.6. Bacillus velezensis and Bacillus subtilis Improved Plant Disease Resistance

In the case of *Botrytis cinerea* inoculated with PDA mycelia plugs, it was treated by spraying *Bacillus velezensis* (BV), *Bacillus subtilis* (BS), and a mixture of *Bacillus velezensis* and *Bacillus subtilis* (BVS). The results showed that spraying BS and BVS significantly inhibited the diameter of the *Botrytis cinerea* on the tomato leaves (*p* < 0.05), and the inhibition rates were 29.28% and 40.99%, respectively. However, for the *Botrytis cinerea* lesion diameter on the tomato leaves, no significant difference was observed among the treatments sprayed with BS (Figure 6C,D). When tomato seedlings were inoculated with spore suspension spray, spraying with BS and BVS significantly reduced the infection rate of *Botrytis cinerea* in the tomato leaves (*p* < 0.05) by 57.69% and 69.23%, respectively, when compared with the control (Figure 6A,B).

## 4. Discussion

In this study, we first found that Se combined with MeJA could reduce the incidence of *Botrytis cinerea* and improve the quality of the tomato. By profiling bacterial communities in the fruit and leaf compartments of tomatoes and isolating antagonistic bacteria from *Botrytis cinerea*, it was found that under MSe2.0 treatment (0.2 mmol/L MeJA + 2.0 mg/kg Se) the diversity of the tomato interleaf bacterial community was increased and the antagonist *Bacillus velezensis* and *Bacillus subtilis* were enriched. Further data from a pot experiment suggested that the two antagonistic bacteria could be used as biocontrol bacteria to inhibit gray mold on tomatoes and could be applied in agricultural production practice. Below we discuss how these findings have advanced our understanding.

The average weight of each fresh tomato fruit was 10 g in the Se-and-MeJA-treatment group, which corresponded to 0.6 g DW, and the average Se concentration was about 332 μg/kg DW (Figure 2B), or about 0.2 μg Se in each treated fruit. The World Health Organization (WHO) recommends a Se intake of 40–60 μg per day for adults. Therefore, the Se concentration in the tomatoes in this study was not enough to pose a risk to human health but can be used as a Se dietary supplement for humans. More and more studies have shown that trace elements such as Se have the potential to improve the quality of fruits and vegetables and to control diseases [25]. Similarly, previous studies showed that when the postharvest fruits were inoculated with *Botrytis cinerea*, the content of malondialdehyde and the activity of LOX decreased when Se was applied to tomato leaves [36]. A similar result was obtained in a study by Durán et al. [37], which reported that after wheat seeds were coated with Na_2_SeO_3_ the endophytes isolated from the wheat can be used to promote plant growth and biological control of *Gaeumannomyces graminis*. Pezzarossa et al. [38] reported that Se application significantly increased the firmness and soluble-solid content of peach fruits. Studies have also shown that MeJA induced the resistance of Chinese bayberry to *Penicillium citrinum* by initiating a defense response [30].

Plants have always been accompanied by beneficial and pathogenic microbes. The metagenomic sequencing of apples indicated that plant pathogens were also found in healthy fruits [39]. Microbes on plants change dynamically with the environment in which the plants grow. Plants form specific microflora in the process of growth, and the specific microflora is susceptible to multiple factors. The traditional process of planting, agronomic measures, cultivar, soil type, and fertilizing has a strong impact on plant microbiota [40,41]. In our study, Se and MeJA changed the composition and structure of the microbial communities in tomato fruits and leaves (Figure 3 and Figure 4). Moreover, the microbial community also changes after the plant is infected by a pathogen. Studies of the microbial communities of mangoes and kiwifruit showed that there were significant changes in microbial communities between healthy and infected samples [42,43]. In addition, the relative abundance of some microbial groups was monitored to judge the decay of sugar beet [44]. 

Emerging evidence indicates that microflora is a key factor in regulating plant resistance to stress and disease. Recent studies have confirmed that plants recruit specific rhizosphere microorganisms and endophytes to enhance their adaptability to salt stress [8]. After wheat was infected by the fungal pathogen *Rhizoctonia solani* AG8 or *Fusarium pseudograminearum*, beneficial microorganisms were recruited, which can inhibit the pathogen or regulate the expression of wheat defense–related genes [14,45]. In our study, after treatment, *Bacilli* was enriched in tomato leaves, which inhibited the growth of *Botrytis cinerea* both in vitro and in plants (Figure 3A). Similarly, Matsumoto et al. [46] isolated *Sphingomonas melonis* from rice seeds, which can produce anthranilic acid and reduce seedling blights and grain rot caused by *Burkholderia plantarii*.

Plant recruitment of beneficial microorganisms may be a conserved survival strategy in the plant kingdom [47]. *Bacillus velezensis* and *Bacillus subtilis* were isolated from tomato leaves and exhibited antagonistic activities to *Botrytis cinerea* (Figure 5). The obstructed growth of *Botrytis cinerea* may be due to nutritional competition between isolated bacteria and *Botrytis cinerea*; the metabolites produced by bacteria in the growth process may inhibit *Botrytis cinerea*. Antagonistic antibacterial can be applied to the prevention and control of gray mold in tomato planting. This finding indicated that *Bacillus velezensis* can stimulate *Pseudomonas stutzeri* rhizosphere to promote plant growth through metabolic interactions [48]. Pre-application of bacterial fertilizer containing *Bacillus subtilis* increased potato yield and persistent scab, which also increased the relative abundance of beneficial bacteria in the rhizosphere [49]. The underlying mechanisms of endophyte-mediated plant resistance should be further explored in order to apply microbial products to sustainable agriculture.

## 5. Conclusions

To sum up, our results showed that the application of Se in soil combined with the foliar spraying of MeJA could reduce *Botrytis cinerea* infection, improve fruit quality, and change the microbial community of tomato fruits and leaves. The enrichment of *Bacilli* in tomato leaves under treatment could enhance the resistance of tomatoes to *Botrytis cinerea*. These findings may benefit our understanding of the new regulation of microorganisms on *Botrytis cinerea*. This may help to garner fundamental clues to develop strategies allowing for more efficient prevention and control of tomato gray mold.

## Figures and Tables

**Figure 1 jof-08-00731-f001:**
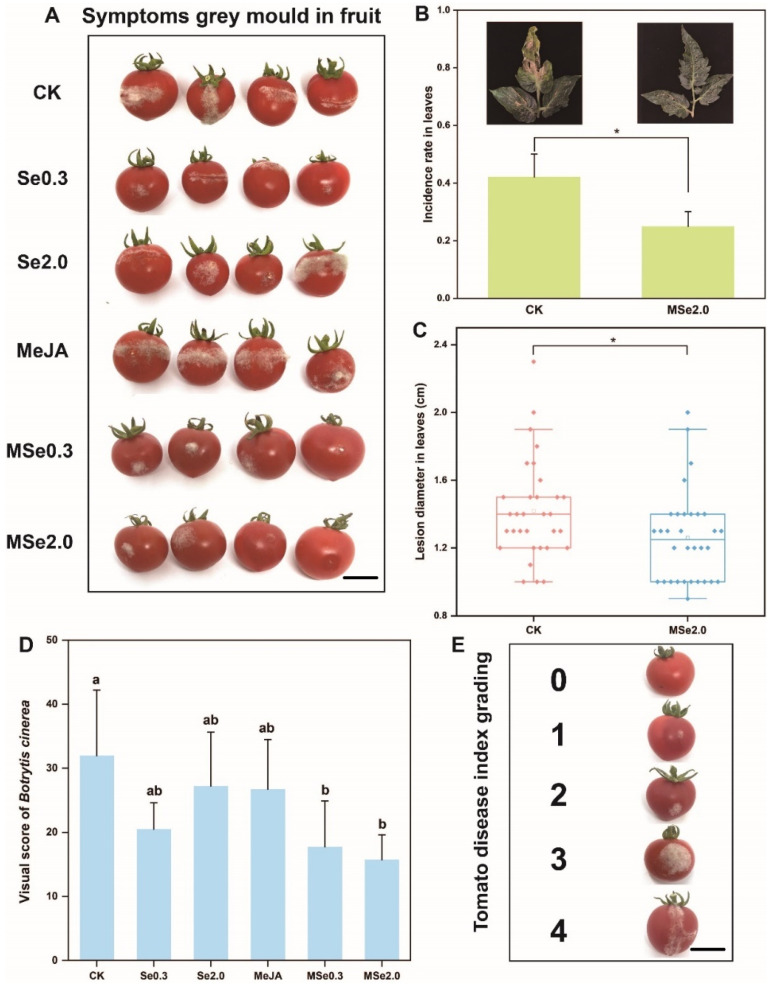
**Effect of Se and MeJA on controlling grey mold in tomatoes.** (**A**) Symptoms of grey mold in fruit. (**B**) Visual score of *Botrytis cinerea*. Grey mold severity was evaluated using a scale of 0 to 4 based on the diameter of the lesions on the tomato surface. (**C**) Box plot shows the average diameter of tomato leaf lesions after inoculation with *Botrytis cinerea.* (**D**) Infection rate of *Botrytis cinerea* in tomato leaves. (**E**) Tomato disease index grading. 0: no spots; 1: 0–3 mm; 2: 3–6 mm; 3: 6–9 mm; 4: >9 mm. Treatments followed by different lowercase letters are statistically different by one-way analysis of variance (ANOVA) (*p* < 0.05). “*” indicates a significant difference between the two treatments by Sudent’s *t* test (*p* < 0.05). Bars = 12 mm.

**Figure 2 jof-08-00731-f002:**
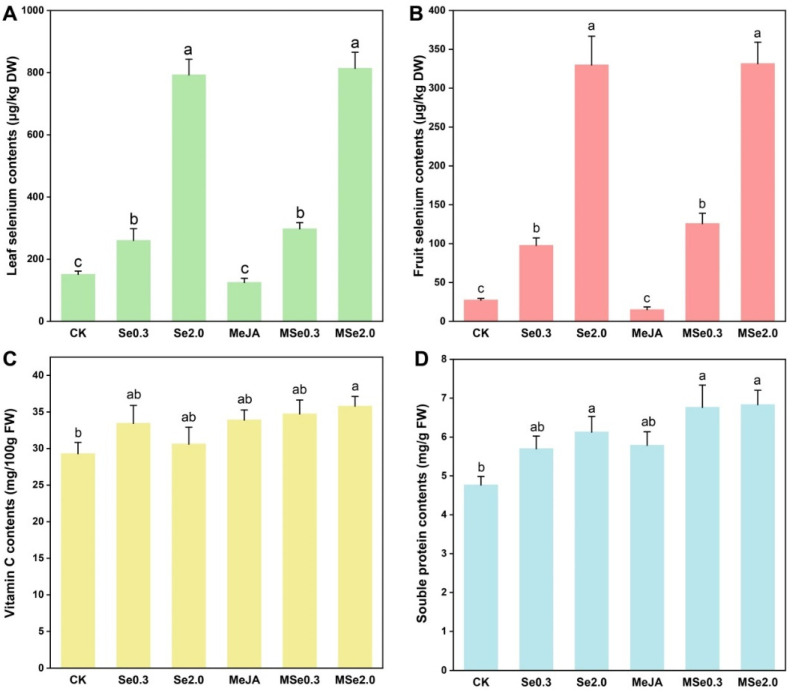
**Effects of Se and MeJA on tomato leaf selenium content.** (**A**), fruit selenium content (**B**), vitamin C content (**C**), and soluble protein content (**D**). Treatments followed by different lowercase letters were statistically different by the one-way analysis of variance (ANOVA) (*p* < 0.05).

**Figure 3 jof-08-00731-f003:**
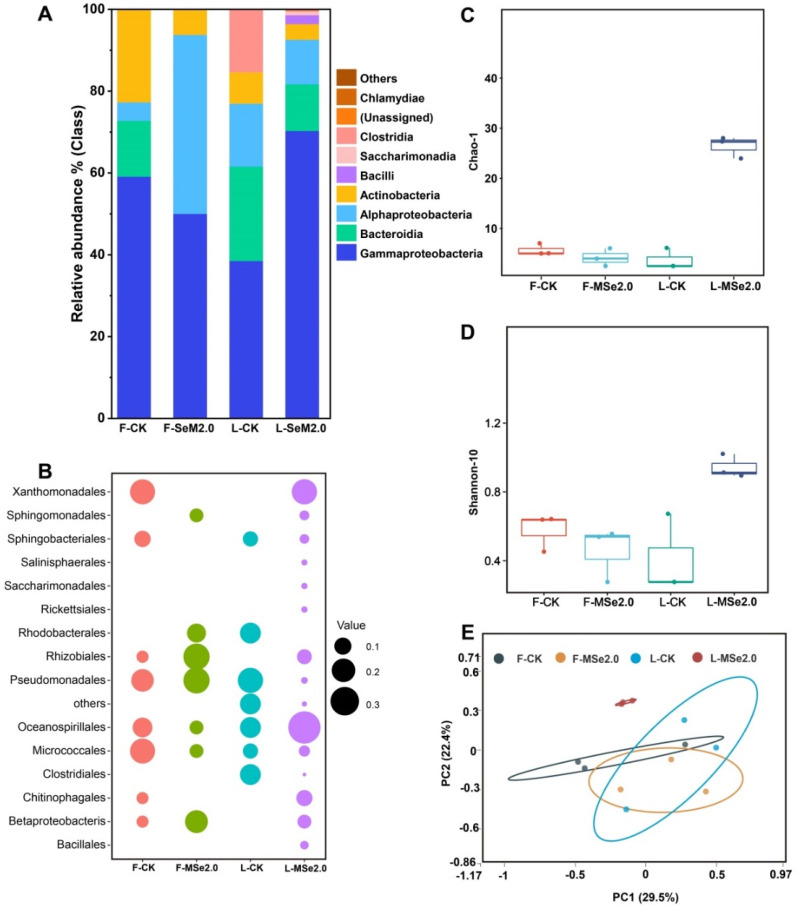
**Effects of Se and MeJA on bacterial community composition and structure in tomato fruit and leaves.** (**A**) Histograms of bacterial species in Se and MeJA treatment and control at class level. (**B**) Bubble chart of bacterial species in Se and MeJA treatment and control at order level. The color code for the bubbles depicts the different treatments, and tomato tissue and bubble size indicate relative abundance of order within the tomato microbiota. (**C**) Evaluation of the microflora richness of tomato by Chao-1 index. (**D**) Evaluation of the microflora diversity of the tomato by Shannon index. (**E**) PCoA of bacterial communities with Bray-Curtis distance in fruit and leaf of tomato. “F” means fruit, “L” means leaf.

**Figure 4 jof-08-00731-f004:**
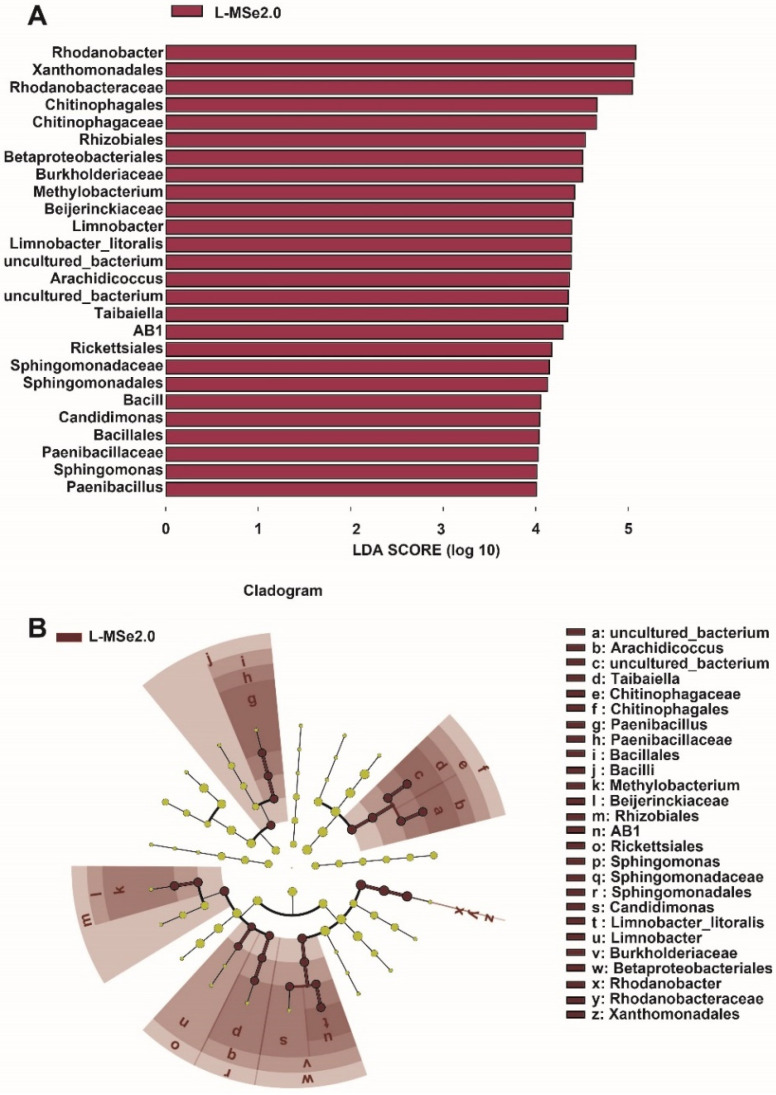
**Different species of microbial communities between MSe2.0 and CK of bacteria on the tomato leaves.** (**A**) LEfSe of phyllosphere bacterial community. (**B**) Cladogram of phyllosphere bacterial community (LDA scores > 4).

**Figure 5 jof-08-00731-f005:**
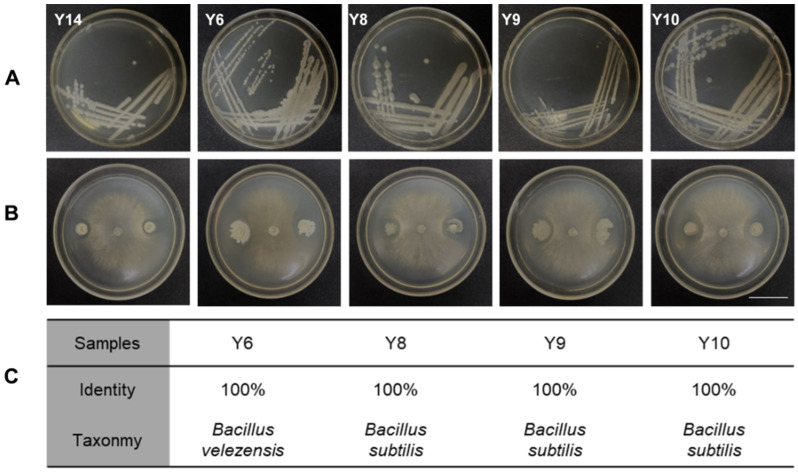
**Isolation of phyllosphere bacteria and their antagonism to *Botrytis cinerea*.** (**A**) Colony characteristic of bacterial isolated from tomato interfoliate. (**B**) Inhibitory effect of tomato interfoliar bacteria on *Botryis cinerea*. (**C**) Annotated results of 4 antagonistic bacteria samples. Bars = 30 mm.

**Figure 6 jof-08-00731-f006:**
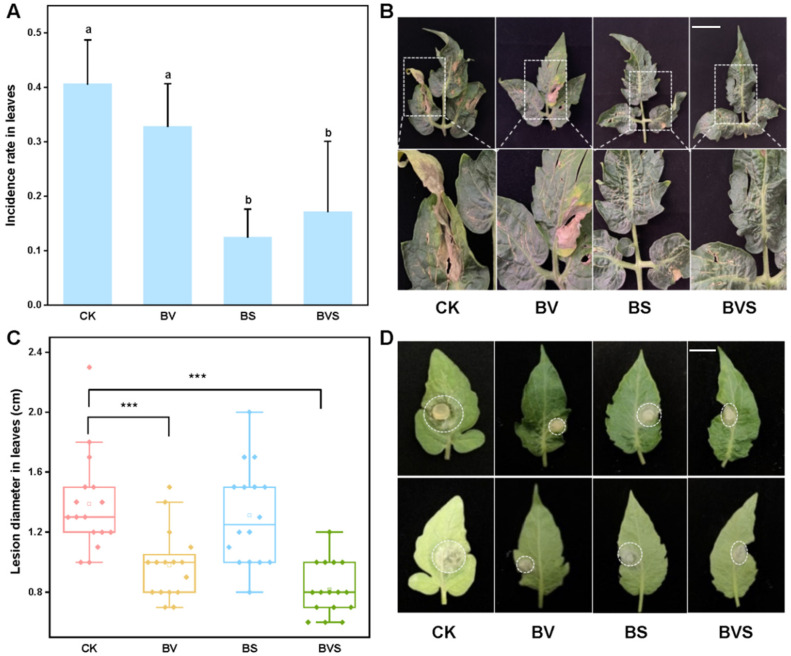
**Effects of *Bacillus velezensis* and *Bacillus subtilis* on tomato disease resistance of *Botrytis cinerea*.** (**A**) Infection rate of *Botrytis cinerea* in tomato leaves. (**B**) Representative incidence of *Botrytis cinerea* tomato leaves that obtained from four beneficial bacteria treatments. Bars=30 mm (**C**) Box plot shows the average diameter of tomato leaf lesions after inoculation with *Botrytis cinerea.* (**D**) Representative disease lesion diameter of *Botrytis cinerea* tomato leaves that obtained from four beneficial bacteria treatments. Bars = 15 mm. Treatments followed by different lowercase letters are statistically different by one-way analysis of variance (ANOVA) (*p* < 0.05). “***” indicates a significant difference between the two treatments by Sudent’s *t* test (*p* < 0.001).

## Data Availability

The data presented in this study are available on request from the corresponding author.

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
