# Peer review of "Selenium Combined with Methyl Jasmonate to Control Tomato Gray Mold by Optimizing Microbial Community Structure in Plants"

_jof, 2022, doi:10.3390/jof8070731_

Round 1
Reviewer 1 Report
The manuscript presents exciting results related to the combined use of methyl jasmonate and selenium compounds to reduce the incidence of Botrytis cinerea and improve tomato quality. However, it is incomplete work and should not be published in the current version. The reasons are as follows:
-The introduction should be improved. Although the importance and background of the manuscript are evident, aspects such as the toxicity of selenium compounds and their excessive use should be delved into. The work of MacFarquhar et al. (10.1001/archinternmed.2009.495) evidence that toxic concentrations of selenium lead to widespread bud formation that can be dangerous for human and animal consumption. In addition, multiple literature references have demonstrated the poisonous nature of many selenium compounds, most notably alkaline diseases in cattle raised in some regions of the American Great Plains and acute and chronic selenosis.
-To reduce the manuscript's coincidence with other references in the literature, modify the text in lines 95-97, 171-173, 184-187. The information with the matching reference is listed below:
"The characteristics of soil used for pot experiments were: organic matter 13.58 g/kg, pH 5.98, available hydrolysis N 38.73 mg/kg, available P 48.33g/kg, available K 225.66 mg/kg, and total Se 0.019 mg/kg".
Reference: https://doi.org/10.1016/j.envpol.2020.114827
"Genomic DNA was extracted and purified from leaf and fruit of tomato samples by using the MOBIO PowerSoil DNA Isolation Kit (MOBIOLaboratories, Carlsbad, CA, USA) following the manufacturer’s recommendations."
Reference: https://www.nature.com/articles/s41396-021-00974-2
"PCoA was performed using the “vegan” package in R based on the Bray-Curtis dissimilarity. Significant differences in the relative abundances of bacterial taxa between treatments were identified using the linear discriminant analysis (LDA) effect size (LEfSe)."
Reference: https://doi.org/10.1016/j.soilbio.2020.108113
-The use of metabolomics and genomics for this kind of studies has grown significantly in recent years. The authors could carry out metabolomics studies using Ultra-High Performance Liquid Chromatography coupled to High-Resolution Mass Spectrometry. Authors must use a protocol similar to that of Tsalgatidou et al. (https://www.mdpi.com/2076-2607 /10/2/399/htm). The latter, through genomic and metabolomic analysis, identified the production of surfactin, fengycin, and mojavensin A, as relevant metabolites due to their antagonistic activity and of bacillaene compounds, which, apart from their antimicrobial activity, accelerate the production of biofilm for better self-protection of bacterial cells for a successful ex vivo colonization and thus, combat possible competing microorganisms.
Reviewer 2 Report
The manuscript entitled “Selenium combined with methyl jasmonate to control tomato gray mold by optimizing microbial community structure in plants” has focused on the effect of selenium and methyl jasmonate application on microbial structure to control tomato gray mold. The idea is novel and is suitable for publication in the journal. English has to be improved.
Specific comments:
Line 20: remove extra dot
Line 42: unsuitable expression “drug resistance”
Line 48: “and resist pathogenic bacteria [9]” unrelated sentence since the manuscript is not talking about pathogenic bacteria
Line 58- 60: what is the relation between the manuscript and biochar? Remove the sentence.
Line 63: add “and” before “enhance”
Line 102-103: The dose of Se was o.3 mg/kg Se and 2.0 mg/kg Se. (Ref.?) or explain why did you choose those doses?
Line 109 and line 119: tomato leaves disease?
Line 128: The experiment chose 20 fruits for each repeat. (English)
Line 109, 119, 122 and 129: All are about determination of disease incidence. Repeating
Line 129: Why did the authors use two different methods to inoculate plants of the same pathogen? Hyphae infection and Spore infection. Justify
Line 142: It is a sentence without a verb. Sentence structure
Line 154: were diluted to 10-1 × 10-5 concentration gradient. Did you mean 10-1 -10-5 ?
Line 159: correct “the 15 strains of leaves isolated by plate dilution” 15 bacterial strains?
Line 215-221: There is no mention of (Fig. 2A) in the text and no mention of that the measures of selenium content was in µg/kg DW. Check throughout the manuscript.
Line 237- 244: no mention of Fig. 3 (A and B) in the text.
Line 285: “inhibition zone” not “isolation zone”. Correct the expression
Line 285-287: The sentence should be in the discussion section.
Line 299-308: No mention of Fig. (B and D) in the text.
Line 285: replace “cultured” by “incubated”
Line 336: put the complete name of MDA. Malondialdehyde
Line 349: change the expression “On the one hand”.
Line 372: correct “in of”.
Line 380-382: correct the sentence
Line 384: correct the sentence
Reviewer 3 Report
I would like to see some data in abstract as well as rewrite the conclusion of your results in the end of abstract
some details should be write in M&M for example the study of bioagents ion disease incidence did not write in M&M
what the bar on the figures
why the discussion is divided to sections
Round 2
Reviewer 1 Report
The manuscript was improved according to the comments. It could be published in the present form.